# Synthesis of CNT@CoS/NiCo Layered Double Hydroxides with Hollow Nanocages to Enhance Supercapacitors Performance

**DOI:** 10.3390/nano12193509

**Published:** 2022-10-07

**Authors:** Xiaoming Yue, Zihua Chen, Cuicui Xiao, Guohao Song, Shuangquan Zhang, Hu He

**Affiliations:** 1Key Laboratory of Coal Processing and Efficient Utilization (Ministry of Education), School of Chemical Engineering and Technology, China University of Mining and Technology, Xuzhou 221116, China; 2School of Resources and Geosciences, China University of Mining and Technology, Xuzhou 221116, China

**Keywords:** supercapacitor, CNT, NiCo-LDH, nanocage

## Abstract

One of the key factors to improve electrochemical properties is to find exceptional electrode materials. In this work, the nickel-cobalt layered double hydroxide (CNT@CoS/NiCo-LDH) with the structure of a hollow nanocage was prepared by etching CNT@CoS with zeolitic imidazolate framework-67 (ZIF-67) as a template. The results show that the addition of nickel has a great influence on the structure, morphology and chemical properties of materials. The prepared material CNT@CoS/NiCo-LDH-100 (C@CS/NCL-100) inherited the rhombic dodecahedral shape of ZIF-67 well and the CNTs were evenly interspersed among the rhombic dodecahedrons. The presence of CNTs improved the conductivity and surface area of the samples. The C@CS/NCL-100 demonstrates a high specific capacitance of 2794.6 F·g^−1^ at 1 A·g^−1^. Furthermore, as an assemble device, the device of C@CS/NCL-100 as a positive electrode exhibits a relatively high-energy density of 35.64 Wh·kg^−1^ at a power density of 750 W·kg^−1^ Further, even at the high-power density of 3750 W·kg^−1^, the energy density can still retain 26.38 Wh·kg^−1^. Hence, the superior performance of C@CS/NCL-100 can be ascribed to the synergy among CNTs, CoS and NiCo LDH, as well as the excellent three-dimensional structure obtained by used ZIF-67 as a template.

## 1. Introduction

Environmental pollution and fossil energy depletion are pressing problems in the international world. A new type of energy conversion and storage devices emerge as the times require [1]. Compared with traditional energy storage devices such as capacitors and batteries, supercapacitors have the advantages of high-power density, fast charge–discharge speed and long cycle life, etc. [2]. They have become one of the hot topics in scientific research and have been widely used in the field of pure electric vehicles and green energy storage and transformation [3,4,5]. According to the different energy storage mechanisms, supercapacitors can be divided into two types: double layer capacitors (EDLC) and pseudo-capacitors [6,7].

EDLCs rely on the process of charge adsorption and desorption on electrode materials’ surface to achieve energy storage and conversion [8]. Carbon materials, for instance, active carbon, are typical materials used in commercial EDLC for their excellent porous structures [9]. In recent years, CNTs have become a powerful candidate to enhance conductivity and stability for electrode materials because of their unique physical structure and excellent electrochemical properties [10,11]. At present, metal–organic framework materials (MOFs) with the advantages of network structure, high porosity and a large specific surface area have displayed great promise in their employment as sacrificial templates and metal precursors [12]. MOFs composites can effectively combine the advantages of MOFs and other functional materials so that new physicochemical properties and improved properties can be received, which cannot be obtained from a single component. They have been widely employed as sacrificial templates for the synthesis of electrode materials, including metal oxides or hydroxides, porous carbons and others, as well as catalysts and sensors fields [13]. Sun et al. [14] prepared a corrugated-layered structure metal organic framework (MOF)/carbon nanotube composite by the solvothermal method. The corrugated-layered structure of MOF(Ni) offered sufficient electrolyte storage space and a fast diffusion channel, and 1D CNTs offered a good electrical conductivity. MOF(Ni)/CNT not only has the highest specific capacitance, but also possesses an outstanding rate capacity (88.6% retention at 10 A·g^−1^). Nevertheless, carbon-based materials have low specific capacitance due to their charge storage mechanism of electrostatic attraction. Pseudo-capacitive electrode materials store charge through rapid and reversible surface redox reactions and have considerable capacitance values [15,16]. One of the most important means to improve the performance of supercapacitors is to develop and research new electrode materials. Transition metal compounds have been considered as ideal electrode materials for pseudo-capacitors owing to their advantages such as higher specific capacitance and power density [17], etc. However, pseudo-capacitors also have drawbacks, for instance low cycle life and poor rate performance, which limit their practical application [18,19].

Metal hydroxides, especially NiCo-LDHs, are considered promising materials because of their tunable composition and high theoretical capacitance [20]. The general chemical formula of LDH is [M^2+^_1−-x_M_x_^3+^(OH)_2_]^x+^(A^n−^)_x/n_·mH_2_O, and its structural model is shown in Appendix A; M^2+^ and M^3+^ represent divalent and trivalent metal cations, respectively. The anion of A^n−^ is adsorbed between layers to maintain electroneutrality. However, the ultrathin layered structure frequently contributes to capacitance degradation and rapid structural collapse at high current density, which affects their electrochemical properties [21]. Coupling compounds with different advantages or constructing reasonable structures are effective ways to improve electrochemical performance [22]. Zeolitic imidazole frameworks (ZIFs), with the characteristics of diverse structure, high surface area, controllable synthesis and good chemical and thermal stability, have been considered as bifunctional materials with both sacrificial templates and metal precursors toward the construction of hollow structures [23]. Liu et al. [24] synthesized tri-metal hydroxide using nickel foam as a substrate and Co-MOF as a precursor, and successfully synthesized Ni_3_MnCo@Co_9_S_8_-QDs/NF nanocomposites with 492.1 mAh·g^−1^ of electric quantity at a 1 A·g^−1^ current density. After charging and discharging 10,000 cycles at this current density, the capacitance retention rate is still up to 90.4%, showing excellent cycle performance. Li et al. [25] prepared an exquisite hierarchical structure of CoS_2_ flake with the two-dimensional MoS_2_ lamella (MoS_2_@CoS_2_), using flake-like ZIF-67 as a precursor through multiple calcination and one-step vulcanization. The inserted 2D cross-linked MoS_2_ multilayers could effectively prevent the interlayer expansion and structural collapse, and then especially ensure the electrochemical performance and morphological stability of the transition metal sulfides. MoS_2_@CoS_2_ reveals an extraordinary specific capacitance of 950 F·g^−1^ at 1 A·g^−^^1^ and an appreciable cycle life with a capacitance retention of 94.6% (10000 cycles at 10 A·g^−1^). In addition, compounding with carbon materials is also one of the methods to improve the properties of samples. The conductivity of materials can be significantly improved by modified carbon, and the volume change in the charging and discharging process can be buffered [26]. Rajesh et al. [27] prepared NiCo-LDH@Ag NWs/g-C_3_N_4_ composite materials by hydrothermal and ultrasonic methods to optimize the structure of nickel-cobalt hydroxide. Experiments show that the 3D nanostructure composites NiCo-LDH@Ag NWs/g-C_3_N_4_ have higher specific capacitance (848.5 F·g^−1^ at a current density of 1 A·g^−1^) than other materials in the paper, and the cycle performance has also been significantly improved (93.5% after 2000 cycles at a current density of 5 A·g^−1^).

In this paper, the material of CNT@CoS/NiCo layered double hydroxide (CNT@CoS/NiCo-LDH) was prepared by the simple etching precipitation method. The three-dimensional structure of CNT@CoS/NiCo-LDH is attributed to the sacrificial template of ZIF-67. The existence of CNTs can enhance the conductivity of the materials and it is beneficial to the structural stability. Moreover, the first vulcanization can further improve the electrochemical performance through the synergistic effect between sulfide and hydroxide. Thus, through structure optimization, CNT@CoS/NiCo-LDH composites can be used as an electrode material with excellent electrochemical properties.

## 2. Materials and Methods

### 2.1. Preparation of CNT@CoS Precursor

A total of 20 mg of carboxylated multi-walled carbon nanotubes (CNTs) was added into 30 mL of methanol with 1 mmol of cobalt nitrate hexahydrate (AR, Co(NO_3_)_2_·6H_2_O, 291 mg) and stirred for 10 min, followed by adding 2-methylimidazole (8 mmol) into the solution for another 2 h. The resulting solution was allowed to stand at room temperature for 24 h to form CNT/ZIF-67. Then, the vulcanizing agent of thioacetamide (TAA, 24 mg) was added into the mixture and refluxed in a water bath at 80 °C for 1.5 h. The suspension was centrifuged to obtain solid precipitate. The CNT@CoS precursor was received by drying the solid precipitate at 60 °C for 12 h under vacuum condition.

### 2.2. Preparation of CNT@CoS/NiCo-LDH

In a typical procedure, the previously obtained precursor of CNT@CoS (1 mmol Co^2+^) was dispersed in 20 mL of ethanol with ultrasound for 10 min to form a suspension. 100 mg (0.34 mmol) nickel nitrate hexahydrate (AR, Ni (NO_3_)_2_·6H_2_O) pre-dissolved by 10 mL of ethanol was added into the previous suspension and kept for 3 h at 30 °C. Then, the resulting mixing was allowed to stand at room temperature for 24 h. Finally, the sample of C@CS/NCL-100 was obtained after centrifuging and drying at 60 °C under vacuum overnight.

In order to determine the influence of different amounts of nickel source on the materials’ properties, four additional amounts of Ni (NO_3_)_2_·6H_2_O were designed: 50 mg (0.17 mmol), 100 mg (0.34 mmol), 150 mg (0.51 mmol) and 200 mg (0.68 mmol), named C@CS/NCL-50, C@CS/NCL-100, C@CS/NCL-150 and C@CS/NCL-200, and the atomic ratio of Ni and Co in the materials are 1:0.17, 1:0.34, 1:0.51 and 1:0.68, respectively.

### 2.3. Characterization

The crystal structure and surface valence of the materials were obtained by X-ray diffractometer (XRD, D8 Advance, Bruker Co., Germany) using Cu K_α_ radiation (λ = 1.5418 Å) in the 2θ range of 0–90^o^, and X-ray photoelectron spectroscopy (XPS, ESCALAB 250Xi, Thermo Scientific Co., USA) using a monochrome A1K_α_ source (hν = 1486.6 eV). The morphologies of the materials were analyzed with a transition electron microscope (TEM, Tecnai G2 F20, 200 kV, FEI Co., USA) and scanning electron microscope (SEM, Quanta250, acceleration voltage 20 kV, FEI Co., USA). The specific surface area and pore-size distribution of the samples were also acquired by a nitrogen desorption test (BET, Quantachrome Co., USA).

### 2.4. Electrochemical Measurements

The electrochemical properties of the materials were tested in a three-electrode system. The anodes were composed of active material, acetylene black and polytetrafluoroethylene (PTFE) at a mass ratio of 8:1:1. The prepared electrode, Hg/HgO and Pt foil were used as a working electrode, reference electrode and counter electrode, respectively, with 6 M of KOH aqueous solution as an electrolyte. The galvanostatic charge–discharge (GCD) tests were performed using a CT-3008-5V battery detection system (Shenzhen, Neware). Cyclic voltammetry (CV) at different sweep speeds (5, 10, 20, 50, 100 mV·s^−1^) in the voltage range of 0–0.5 V and electrochemical impedance spectroscopy (EIS, 10^5^~10^−2^ Hz, AC voltage amplitude 10 mV) tests were carried on a CHI604E electrochemical workstation (Shanghai, Chenhua).

In order to appraise the practical value of the samples, the electrochemical analyses were carried out in a two-electrode system that was assembled with the prepared samples as a positive electrode, activated carbon (AC) as a negative electrode and 6 M of aqueous solution KOH as an electrolyte. The slurry was coated on two pieces of nickel foam (1 × 1 cm^2^).

## 3. Results and Discussion

### 3.1. Characterization of the Samples

Figure 1 illustrates the synthesis procedure of CNT@CoS/NiCo-LDH. Firstly, ZIF-67 with a rhombic dodecahedral structure was formed on the CNTs. Then, Co^2+^ in the ZIF-67 was reacted with S^2−^ to obtain CoS. In this process, the hollow rhombic dodecahedral structure was obtained because a part of ZIF-67 was decomposed by vulcanization. Finally, after adding the Ni (NO_3_)_2_, the NiCo-LDH was generated on the surface of CoS or ZIF-67 by the reaction of Co^2+^ with Ni^2+^, but the CoS was not completely decomposed to maintain the rhombic dodecahedral structure. Therefore, the existing forms of element Co in the material include CoS and NiCo-LDH. By controlling the addition ratio of Ni (NO_3_)_2_, the kinetic balance of acid etching and precipitation reaction can be adjusted and optimized, which is very important for the ration preparation of hollow NiCo-LDH [28].

In order to investigate the morphology of CNT@CoS/NiCo-LDH, SEM tests were carried out. As depicted in Figure 2a, ZIF-67 has a typical rhombic dodecahedral structure with a diameter of about 500 nm and a smooth surface. After the CNTs were added, the morphology and structure of CNT/ZIF-67 remained basically unchanged, except for the uniform interpenetration of the CNTs in Appendix A. Unlike ZIF-67, there are some defects on the surface of CNT@CoS (Figure 2b) with a few rough places on the surface. The arrays of nanosheets assembled in the polyhedral shell successfully and inherited the morphology of the ZIF-67 templates in the sample of C@CS/NCL-50 (Figure 2c,d) and C@CS/NCL-100 (Figure 2e,f). The CNTs in the samples of CNT@CoS, C@CS/NCL-50 and C@CS/NCL-100 are inserted into the three-dimensional blocks to maintain the stable rhomboidal dodecahedron structure. The specific surface area and porosity of the materials can be raised effectively for a multistage structure, which enables a subsequent increase in active sites on the surface of CNT@CoS/NiCo-LDH. Therefore, due to the composite structure of the materials, the penetration of electrolyte ions and the transport speed of electrons and ions can be further improved. Moreover, from Figure 2c–h, it can be observed that the materials prepared with different masses of nickel nitrate have different surface morphologies. Nickel ions mainly react with cobalt ions in ZIF-67 to form NiCo-LDH and precipitate on the surface of CNT@CoS. Figure 2d,e exhibit that C@CS/NCL-50 and C@CS/NCL-100 have the rhomboidal dodecahedron structure with a rough surface and surrounding CNTs. Comparatively, the composite materials in Figure 2g,h did not maintain the dodecahedron structure of ZIF-67, as the nickel nitrate was added excessively, but formed the lamellar structure, and many sheet structures of NiCo-LDH (Appendix A) agglomerated obviously. It is speculated that the reaction speed of nickel and cobalt ions would accelerate if the concentration of nickel was excessive. While the precursor decomposes, the NiCo-LDH has insufficient time to precipitate, which results in samples that cannot maintain the rhomboidal dodecahedron structure. Therefore, the amounts of nickel nitrate that are added play an important role in the morphologies of the materials.

The samples were further characterized by TEM, as shown in Figure 3. The ZIF-67 (Figure 3a) and CNT/ZIF-67 (Appendix A) both have a solid dodecahedral structure. Figure 3b shows that CNT@CoS has a hollow dodecahedral structure after the vulcanization of ZIF-67 in the material. This is attributed to S^2−^ reacting with the dissolved Co^2+^ on the surface of ZIF-67 to form CoS around the template, but the core of ZIF-67 was gradually consumed until it disappeared completely to form the hollow structure [29]. Figure 3d shows that C@CS/NCL-100 is a hollow nanocage with uniform size. Appendix A shows that Ni, Co and S exist in C@CS/NCL-100. The HRTEM image (Appendix A d) shows that the interplanar spacings of 0.18, 0.21 and 0.22 nm correspond to the (100), (101) and (002) diffraction planes of NiCo-LDH [30]. It can also be seen that part of the CNTs were tightly coated on the surface of the precursors in Figure 3c. Figure 3e,f exhibit that the materials are agglomerated together due to the excessive addition of nickel ions, which is consistent with SEM.

Figure 4 shows the typical X-ray diffraction patterns of CNT@CoS and CNT@CoS/NiCo-LDH composites. The characteristic diffraction peaks of CNT@CoS/NiCo-LDH at 10.9°, 22.2°, 33.5° and 60.3° correspond to (003), (006), (009) and (110) crystal planes, respectively, indicating that the NiCo-LDH has been synthesized successfully [21]. There is an obvious peak at 26.2°, which may be due to the influence of CNTs or CoS in the material of CNT@CoS/NiCo-LDH. In addition, in view of its amorphous structure, the characteristic diffraction peak of CoS is relatively less obvious. Moreover, the CNTs have no obvious diffraction peak, which can be ascribed to the small content of CNTs in the materials of CNT@CoS and CNT@CoS/NiCo-LDH.

CNT@CoS and CNT@CoS/NiCo-LDH have similar type-IV nitrogen adsorption–desorption isotherms in Figure 5a. The hysteresis loops can be clearly observed, which means that there is a considerable amount of mesopores in the samples. It can be seen from Figure 5b that the pore-size curves of two samples have a similar distribution trend. They are both widely distributed in the range of 25 nm. As can be seen from the illustrations, the micropore rate of the CNT@CoS/NiCo-LDH is greater than that of CNT@CoS, and the corresponding mesopore rate decreases. A proper ratio of meso-micropores can not only reduce the internal resistance of the material, but also bestow it a higher specific capacitance. Compared with CNT@CoS (59.1 m^2^·g^−1^), the specific surface area of CNT@CoS/NiCo-LDH is significantly increased (81.3 m^2^·g^−1^, respectively). The specific surface area was calculated with the Brunauer–Emmett–Teller (BET) method, and the pore-size distribution was obtained based on the Density Functional Theory (DFT) method.

XPS was carried out to further evaluate the element composition information of CNT@CoS/NiCo-LDH (Figure 6). Figure 6b shows that Ni 2p has two strong diffraction peaks at Ni 2p_1/2_ (873.8 eV) and Ni 2p_3/2_ (856.2 eV) [31], with two vibration satellite peaks at 861.5 and 880.2 eV, indicating the existence of Ni^2+^ in the sample. As depicted in Figure 6c, the Co 2p XPS spectrum is divided into two main diffraction peaks located at 780.8 and 796.8 eV, which can be assigned to Co2p_1/2_ and Co2p_3/2_, respectively [32], and proved the successful synthesis of NiCo-LDH. In addition, the diffraction peaks at 778.2 and 802.4 eV exhibited that Co^3+^ is also present, further indicating that Co^2+^ and Co^3+^ coexist in CNT@CoS/NiCo-LDH. Based on this, a possible generation mechanism is speculated. The ZIF-67 particles dispersed in Ni (NO_3_)_2_ ethanol solution were etched gradually by protons that were generated from the hydrolysis of Ni^2+^ ions and released Co^2+^ ions. Partial Co^2+^ ions were oxidized by dissolved O_2_ and NO_3_^−^ ions in the solution to form Co^3+^. Then, Co^2+^/Co^3+^ and Ni^2+^ were coprecipitated to form NiCo-LDHs [33]. The supposition is consistent with the XPS results.

The S 2p peaks at 161.7 eV and 168.9 eV can be attributed to the S 2p_3/2_ and S 2p_1/2_ orbitals of CoS_x_ [34,35], respectively, in Figure 6d. In Figure 6e, C 1s uncovers four typical binding energy peaks at 284.6, 285.2, 286.3 and 288.5 eV, corresponding to graphite carbon phase (sp^2^ C-C), sp^3^ hybrid carbon (sp^3^ C-C), sp^3^ C-N and C-O, respectively [36,37,38]. The peaks of O 1s (Figure 6f) of the samples at 531.8 eV and 531.15 eV can be assigned to O-C and O-Co bonds, respectively [39]. In summary, the XPS spectrum demonstrates the presence of Ni, Co, O, S and C in CNT@CoS/NiCo-LDH composites.

### 3.2. Electrochemical Measurements

The GCD curves of CNT@CoS/NiCo-LDH composites at 1–5 A·g^−1^ within the potential window of 0–0.5 V are presented in Figure 7a. The specific capacitance is calculated according to Equation (1):C = IΔt/mΔV,(1)
where C (F/g) represents the specific capacitance, I (A) is the charge and discharge current, Δt (s) implies the discharge time, m (g) signifies the mass of the active material, and ΔV (V) means the voltage range.

As can be seen from the GCD diagram (Figure 7a) of the current density of 1 A·g^−1^, a pair of platforms appeared during the charge–discharge process, which indicates its pseudocapacitive nature. According to Equation (1), the specific capacitance value of the electrode material obtained under the condition of 100 mg is the highest of 2794.6 F·g^−1^ at 1 A·g^−1^, as seen in Table 1.

As shown in Table 1 and Figure 7b, C@CS/NCL-100 has the highest specific capacitance at any current density. The unique structure of C@CS/NCL-100 and the synergistic effects between the three components jointly promoted excellent performance. First, the hollow structure provided a larger surface area and accelerated ion diffusion. Second, the conductivity of the materials was enhanced by the insertion of CNTs and the introduction of sulfide. Third, the surface utilization of the active material was been improved by adding appropriate amount of nickel source.

It can be seen from Figure 7c–f that a pair of representative redox peaks is displayed on the CV curves at different sweep rates (5, 10, 20, 50, 100 mV·s^−1^). As the scanning rate increased, the oxidation peaks moved toward a larger positive potential, and correspondingly, the reduction peaks also moved toward a more negative potential. The mechanism of electrochemical reaction could be expressed as follows [21,40]:Ni (OH)_2_ + OH^−^⇔NiOOH + H_2_O + e^−^
Co (OH)_2_ + OH^−^⇔CoOOH + H_2_O + e^−^
CoOOH + OH^−^⇔CoO_2_ + H_2_O + e^−^

As a typical “battery-type” material, CNT@CoS/NiCo-LDH usually has high charge storage capacity and slow dynamics behavior because of the Faradaic reaction for Co^2+^/^3+^ and Ni^2+^/^3+^. Observably, the integrated area under the CV curve of C@CS/NCL-100 is larger than others under the same scan rate, further indicating its superior specific capacitance.

In order to further investigate the electrochemical behavior, the surface phenomena and dynamics of the electrodes, an electrochemical impedance spectroscopy (EIS) measurement and analysis were carried out. The Nyquist plots of the materials and the electrical equivalent circuit are presented in Figure 8. The charge transfer resistance (*R*_ct_) can be obtained from the semicircle in the intermediate frequency region. According to the linear slope at the low-frequency region, the Warburg impedance (*W*) can represent the ion diffusion impedance and internal contact resistance inside the electrode. The real axis in the intersection of the high-frequency region demonstrates that the series resistance of the overall system (*R*_s_) value of C@CS/NCL-100 electrode is 0.56 Ω, which is lower than the other three samples. It indicates that the material of C@CS/NCL-100 has faster electron transfer and ion transport compared with C@CS/NCL-50, C@CS/NCL-150 and C@CS/NCL-200. Moreover, the C@CS/NCL-100 has a small semicircular diameter in the high-frequency region and the largest linear slope in the low-frequency region, indicating that C@CS/NCL-100 has small contact resistance and faster electrolyte ion diffusion.

### 3.3. Assembly of All-Solid-State Asymmetric Supercapacitor

The electrochemical analysis in a two-electrode system was carried out to deeply assess the energy storage performance of the C@CS/NCL-100 electrode for potential practical applications, such as the determination and calculation of energy density, power density, cyclic stability, coulombic efficiency and so on. The device was fabricated using C@CS/NCL-100 as a positive electrode with AC as a negative electrode. The mass ratio of two electrode materials was calculated according to the charge balance theory given in Equation (2).
(2)$$\frac{m_{+}}{m_{-}}=\frac{C_{-}V_{-}}{C_{+}V_{+}},$$

As shown in Figure 9a, 0–1.5 V is considered as an ideal working potential window for the GCD profile of the device. The device with C@CS/NCL-100 exhibited the specific capacitance of 114.1, 90.4, 87.0, 86.2 and 84.4 F·g^−^^1^ (the capacity is 47.7, 37.7, 36.3 and 35.2 mAh·g^−1^) at the current density of 1, 2, 3, 4 and 5 A·g^−1^, respectively, which is better than that of CNT@CoS at the same current density (87.9, 76.3, 69.8, 65.0 and 62.7 F·g^−1^, respectively, or 36.6, 31.8, 29.1 and 26.1 mAh·g^−1^, respectively). It can be attributed to the nanocage-like structure of C@CS/NCL-100 prepared with nickel source. The deposition of nanosheets increases the roughness of the nanocage surface. Therefore, the specific surface area and active sites were improved to increase the specific capacitance of the material, comparing with CNT@CoS. It can be judged from Figure 9b that 0–1.5 V is the ideal voltage window for the electrochemical test of the device.

A Ragone plot was regarded as a critical indicator of the supercapacitor properties for practical application. The energy density and power density of the device were calculated according to the following equation:(3)$$E=\frac{1}{7.2}{C\Delta V\;}^{2},$$
(4)$$P=3600\;\frac{E}{\Delta t}$$
where E is the energy density (Wh·kg^−1^), P represents the power density (W·kg^−1^), C refers to the specific capacitance (F·g^−1^), ΔV is the voltage window (V), and Δt is the discharge time (s) [41].

As seen in Figure 9c, the energy density of CNT@CoS is only 27.46 Wh·kg^−1^, while CNT@CoS/NiCo-LDH can be up to 35.64 Wh·kg^−1^ at 1 A·g^−1^. Even at the higher power density of 3750 W·kg^−1^, the device can still be retained at 31.38 Wh·kg^−1^, which has a better performance than the CNT@CoS asymmetric system. From Appendix A, it shows the material of CNT@CoS/NiCo-LDH in the application of supercapacitor has an excellent electrochemical performance of energy density and power density compared with other NiCo-materials [28,42,43,44,45]. 

Moreover, 5000 cycles of charge–discharge tests were conducted at the current density of 1 A·g^−1^ to verify the stability of the device in Figure 9d. The device of CNT@CoS/NiCo-LDH maintains the capacitance retention of 87% (99.3 F·g^−1^) compared with the initiate value after 1000 cycles of charge–discharge and maintains 71.4% (81.5 F·g^−1^) after 5000 cycles, which is better than that of CNT@CoS (51.8%, 45.5 F·g^−1^). This can be attributed to the addition of nickel nitrate, resulting in a more stable layered nanocage structure to enhance the electrochemical properties.

## 4. Conclusions

In summary, a MOF-derived nanocage CNT@CoS/NiCo-LDH composite has been prepared by simple vulcanization before employing etching precipitation methods. The sample of C@CS/NCL-100 maintains a rhomboidal dodecahedron structure of ZIF-67, and CNTs were evenly inserted into the hollow structure. The C@CS/NCL-100 nanocomposites with a unique multistage structure have a larger specific surface area of 81.3 m^2^·g^−1^ than CNT@CoS. Benefiting from the above features, the C@CS/NCL-100 electrode performs a high specific capacitance of 2794.6 F·g^−1^ at 1 A·g^−1^, which is higher than that of CNT@CoS (2106.9 F·g^−1^). After 1000 cycles, the capacity retention was 87%. Furthermore, the device displays a fairly high energy density of 31.38 Wh·kg^−1^ at a power density of 3750 W·kg^−1^. Overall, C@CS/NCL-100 with an excellent rhomboidal dodecahedron structure has a certain competitiveness as the electrode material of supercapacitors.

## Figures and Tables

**Figure 1 nanomaterials-12-03509-f001:**
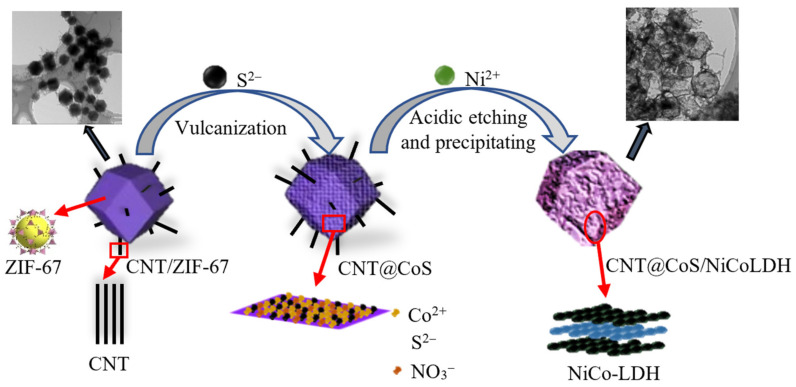
Proposed synthesis pathway of CNT@CoS/NiCo-LDH.

**Figure 2 nanomaterials-12-03509-f002:**
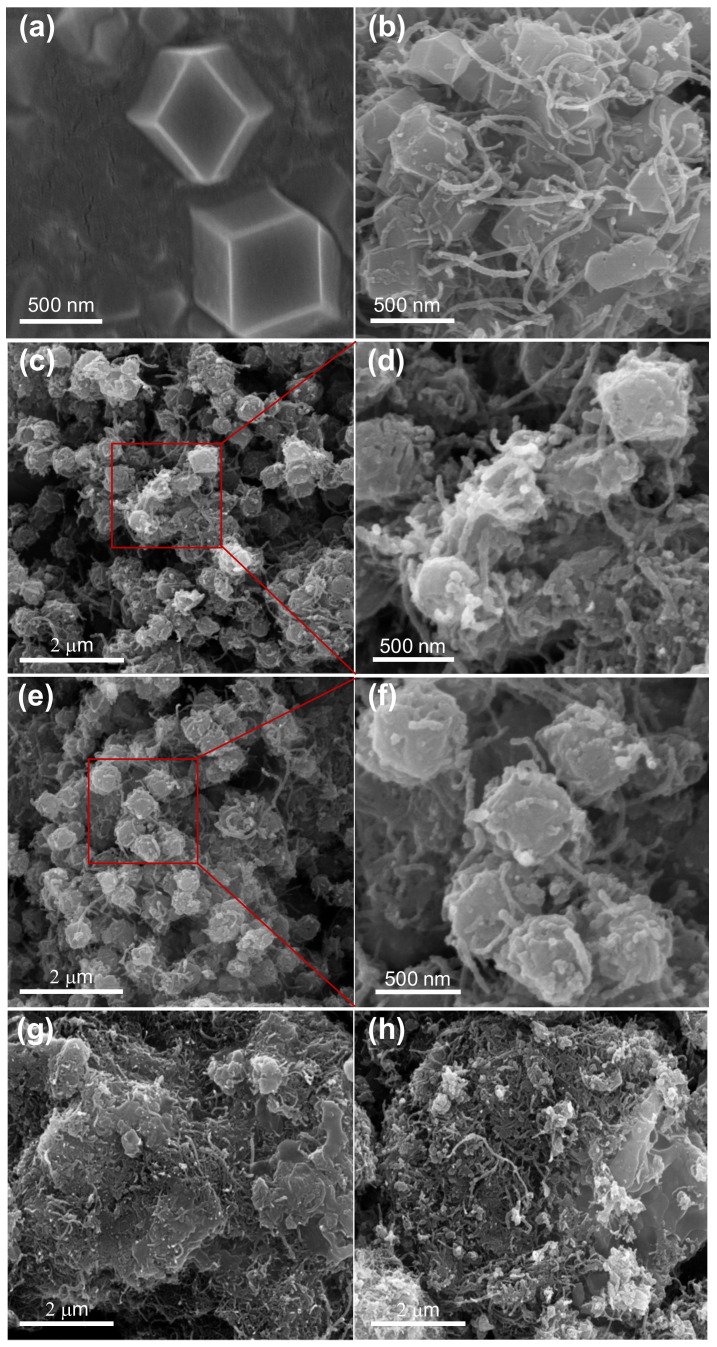
SEM images of (**a**) ZIF-67, (**b**) CNT@CoS, (**c**,**d**) C@CS/NCL-50, (**e**,**f**) C@CS/NCL-100, (**g**) C@CS/NCL-150, (**h**) C@CS/NCL-200.

**Figure 3 nanomaterials-12-03509-f003:**
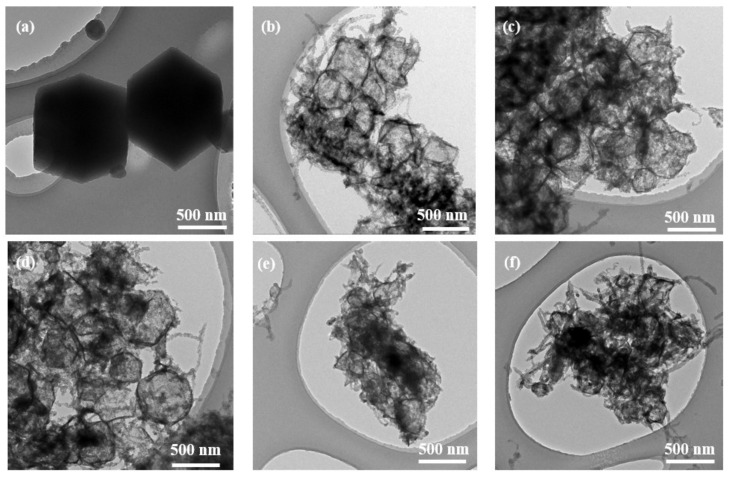
TEM images of (**a**) ZIF-67, (**b**) CNT@CoS, (**c**) C@CS/NCL-50, (**d**) C@CS/NCL-100, (**e**) C@CS/NCL-150, (**f**) C@CS/NCL-200.

**Figure 4 nanomaterials-12-03509-f004:**
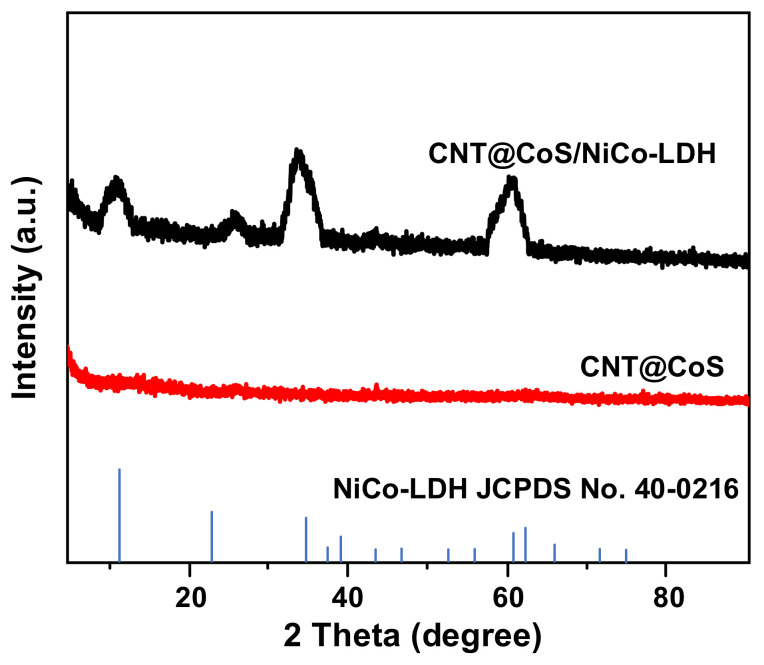
XRD patterns of CNT@CoS and CNT@CoS/NiCo-LDH.

**Figure 5 nanomaterials-12-03509-f005:**
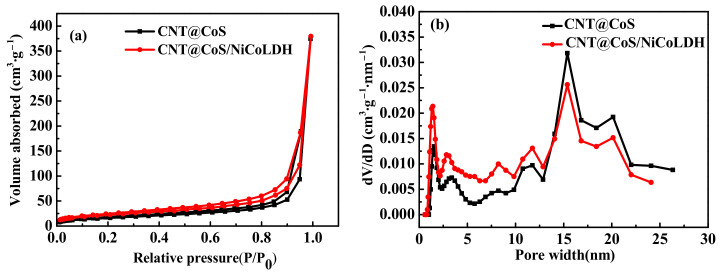
(**a**) N_2_ adsorption–desorption isotherm of samples; (**b**) pore-size distribution curves of the samples.

**Figure 6 nanomaterials-12-03509-f006:**
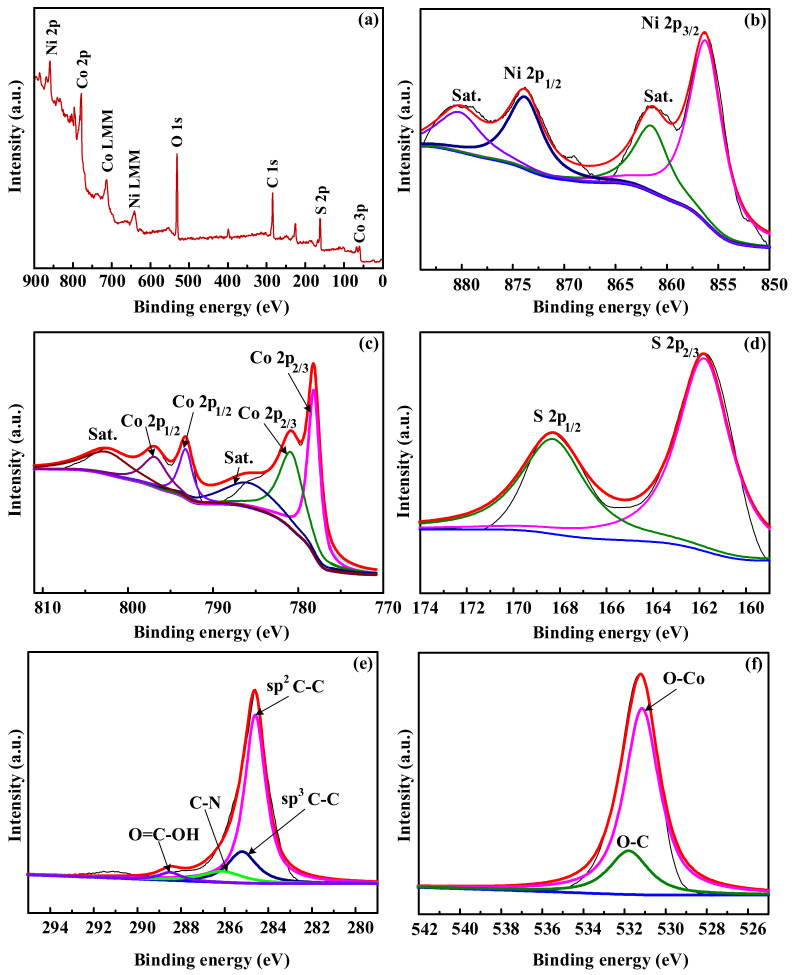
XPS spectra of the elements of CNT@CoS/NiCo-LDH. (**a**) Data survey spectrum of CNT@CoS/NiCo-LDH, (**b**) Ni 2p, (**c**) Co 2p, (**d**) S 2p, (**e**) C 1s, (**f**) O 1s.

**Figure 7 nanomaterials-12-03509-f007:**
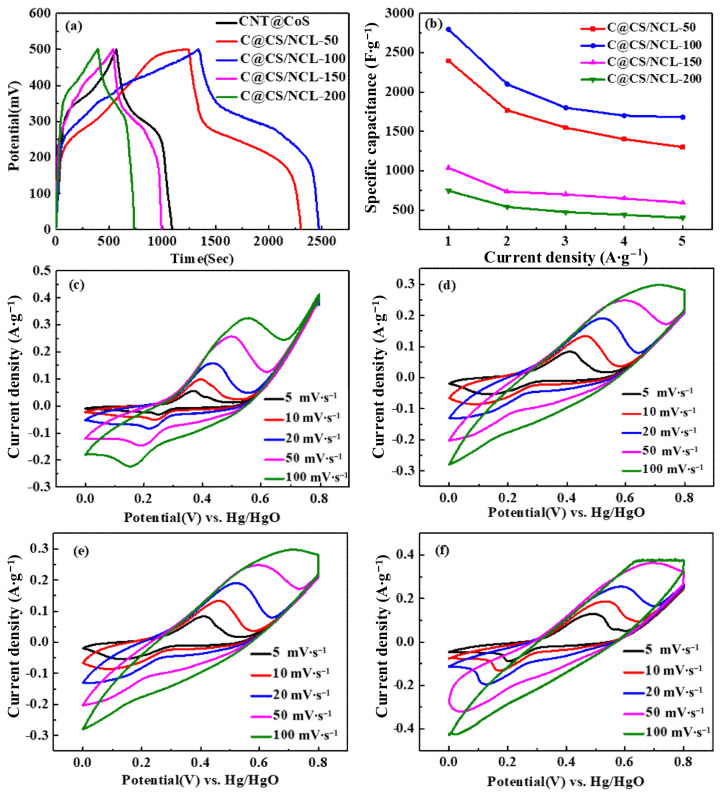
(**a**) Comparation of the GCD curves of the materials’ electrodes with different masses of nickel source, (**b**) their capacitance at different current densities, and CV curves of CNT@CoS/NiCo-LDH at different scan rates; (**c**) C@CS/NCL-50, (**d**) C@CS/NCL-100, (**e**) C@CS/NCL-150, and (**f**) C@CS/NCL-200.

**Figure 8 nanomaterials-12-03509-f008:**
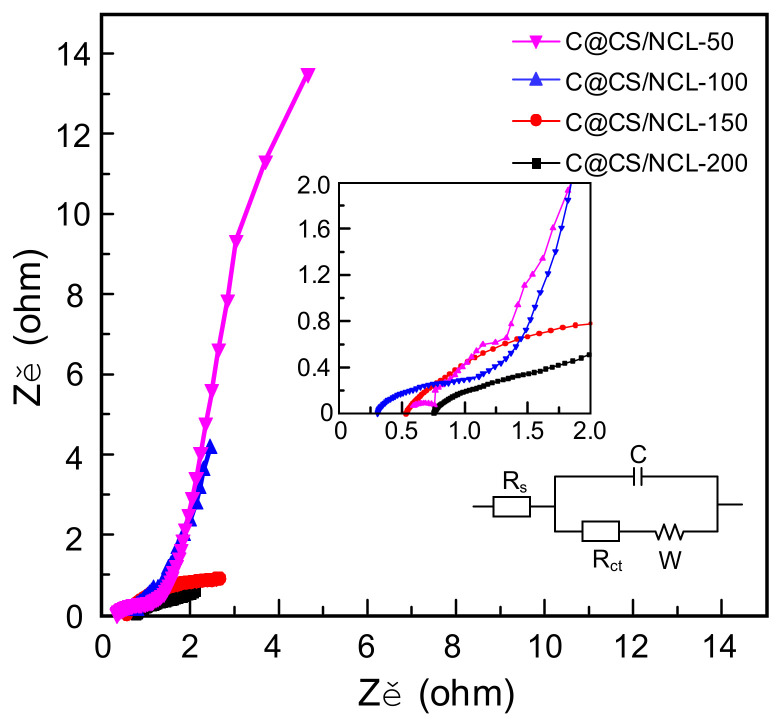
Nyquist plots of C@CS/NCL-50, C@CS/NCL-100, C@CS/NCL-150 and C@CS/NCL-200, respectively.

**Figure 9 nanomaterials-12-03509-f009:**
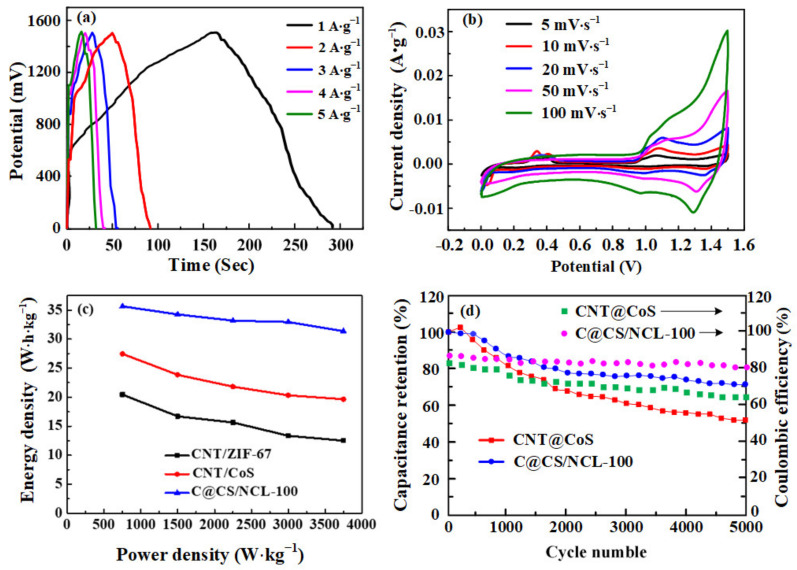
(**a**) GCD programs at different current densities for C@CS/NCL-100; (**b**) CV curves of C@CS/NCL-100; (**c**) Ragone plot of C@CS/NCL-100 and C@CS devices; (**d**) cyclic stability test for C@CS/NCL-100 and CNT@CoS devices at 1 A·g^−1^.

**Table 1 nanomaterials-12-03509-t001:** Specific capacitance values of samples at different current densities.

Current Density (A·g^−1^)	1	2	3	4	5
C@CS/NCL-50	2396.7	1768.1	1547.4	1399.6	1299.7
C@CS/NCL-100	2794.6	2100.5	1800.4	1700.7	1680.0
C@CS/NCL-150	1035.5	733.9	699.0	648.7	592.7
C@CS/NCL-200	746.0	733.9	473.9	439.9	400.0
CNT@CoS	1920.9	1679.8	1550.1	1414.8	1284.4

## Data Availability

The data presented in this study are available on request from the corresponding author.

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
