# Peer review of "Synthesis of CNT@CoS/NiCo Layered Double Hydroxides with Hollow Nanocages to Enhance Supercapacitors Performance"

_nanomaterials, 2022, doi:10.3390/nano12193509_

Round 1
Reviewer 1 Report
Dear Editor,
in this paper, the authors investigate promising systems for supercapacitors electrodes, by coupling properties of Layered Doble Hydroxides (LDHs) with CNTs and the framework of ZIF-67 structure, in order to improve electrochemical performances. Authors proved that this multistage structure can raise surface area, porosity and ultimately active sites of electrodes. Moreover also conductivity can be increased, due to CNTs presence in the composites. Samples were characterized with several experimental techniques ,to understand the mechanisms of formation of composites. Finally, an increasing of electrochemical performance is demonstrated .
The paper has a a fair/good degree of novelty, but very often the sentences in English make the reading lose clarity. Nevertheless, after a thorough syntax check, I can recommend a publication on NM with minor revisions (only a couple of observations are a bit more critical)
Furthermore, I propose to add two plural in the title for the words “hydroxide” and “nanocage” (but this up to the authors): “Synthesis of CNT@CoS/NiCo Layered Double Hydroxides with Hollow Nanocages to Enhance Supercapacitors Performance”
Here is a List of Observations/required corrections:
|
Raw |
Correction / observations |
|
13 |
Please specify: what does it mean 100 mg “of optimal addition”? 100 mg on which solution? Also in “materials and methods” you specify that you add 100 mg of nickel nitrate in a given molarity of Co-based or CoS-based precursor: but.. what is the initial volume of solution in which to dissolve 100 mg of nickel nitrate ? |
|
28 |
Ref 1 is ok, but here people could expect and appreciate also a more general paper on this introductory argument: please add one (may be a good review |
|
30 |
The same of previous raw: Ref 2 is ok, but here people could expect and appreciate also a more general paper on this introductory argument: please add one (may be a good review ) |
|
41 |
Please define MOF acronym, the first time |
|
58 |
The quality of figure S1 is really poor (at least in this file) could you please make some improvement? |
|
63 (and other around) |
You “jump” quickly form MOF to ZIF (and vice versa), and reading results to be not completely clear :could you add (somewhere) a sentence to specify why , inside the family of MOF, you have chosen ZIF67? |
|
73 and 81 (and others) |
please choose a homogeneous style to indicate the thousands |
|
92-93 |
This sentence “sounds” strange: please check (especially “which because” and the verb eems missing) |
|
97-98 |
May be: “can BE used” ? |
|
124-126 |
“Microscope… were conducted” ? please check this sentence |
|
140 |
Please, for the sake of clarity , specify better what is “practical value of the samples” |
|
170-171 |
Please check English of this sentence |
|
174-175 |
Please check English of this sentence : may be you can add something like “to the fact that” |
|
179-180 |
May be a verb “is” I missing: “The concentration of nickel IS excessive” ? |
|
189-191 |
Please check English of this sentence |
|
200 |
A problem with XRD values: id 11.38° is the (003) peak, the 26.2° CANNOT be (006): because, from Bragg’s law, it should be around 22.9. Please try to explain what is the peak at 26.2 ° (or, at least, do not index it as (002) |
|
203-204 |
Another sentence not “sounding” well |
|
216 |
Please explain (for not experts) how you obtain these values of specific surface area from your measurements |
|
227 |
May be, you can change “indicates” with “indicating” |
|
242 |
Before “assigned” perhaps tyou can put “can be assigned…” |
|
249-264 |
Please correct the italic font also in the figure caption) |
|
256 |
Please check this sentence “Table 1 of specific capacitance…” |
|
302 |
perhaps the plural “nanosheets” is better |
|
324 |
A verb “is” could be added after “which”, I think |
Reviewer 2 Report
In this work, ‘Synthesis of CNT@CoS/NiCo Layered Double Hydroxide with Hollow Nanocage to Enhance Supercapacitors Performance, the authors have prepared NiCo-based layered double hydroxide composite with carbon nanotube and CoS nanocages. The composite possing optimum nickel content showed a higher specific capacitance of 2794 F g-1 at 1 A g-1. Further, the assembled asymmetric device delivered an energy density of 35 Wh kg-1. The manuscript, however, requires a majosr revision before publishing in “Nanomaterials”. The authors should consider the following comments and revise the manuscript accordingly.
1. What is the rationale to change the ratio of nickel in the preparation of NiCoLDH. During the formation of NiCo LDH sheets, the atomic ratio of Ni and Co are fixed. If the content of Ni is particularly changed, the excess Ni may be deposited as NiO over the NiCo LDH sheets. From the SEM image of pristine NiCo-LDH (Fig. S3), it is clear that the sheet-like morphology has other impurities on the surface. Alternatively, authors could prepare NiCo-LDH sheets separately for comparison and mix them with other components in different weight ratio to study their role in electrochemical performance. Further, they could highlight their method of preparation that might be optimum.
2. Authors should provide the exact atomic ratio of Ni and Co in the composites.
3. High-resolution SEM images of Fig.2c and 2d should be included as insets to get a clear idea of the morphology.
4. In order to make sure the presence of NiCo LDH in these composites, the TEM elemental mapping data need to be presented. In the optimum composite (C@CS/NCL-100; Fig. 3c), we could not find any NiCo-LDH layers. Better HR-TEM images should be provided.
5. In Fig. 4, the JCPDS file number and standard diffraction pattern of Ni-Co LDH should be included.
6. High-resolution TEM images of optimum (C@CS/NCL-100) composite with lattice fringes are needed to get more evidence (J. Colloid Interface Sci. (2022) 619 75-83; DOI: 10.1016/j.jcis.2022.03.056)
7. XPS survey spectra of the materials should be presented in the supplementary information.
8. A circuit fitting model of impedance spectra should be provided in Fig. 8.
9. The coulombic efficiency of full-cell device looks poor (Fig. 9a). Better electrochemical data should be presented in the revised manuscript.
10. In the cyclic stability experiment (Fig. 9d), the data on coulombic efficiency should be included. Further, the number of cycles is also less. In general, the number of cycles should be more than 5000 cycles for supercapacitors to demonstrate the stability of the device.

Reviewer 3 Report
The current manuscript demonstrates the supercapacitor application of CNt@CoS/NiCo LDH composite. The manuscript has significant data to be published in this journal. However, a few weak points are there in the manuscript, which should be rectified before publication. Therefore, I recommend "Major Revision" of this article. My comments are given below -
1. Fig. 4: Why there is no peak for CNT in the XRD pattern of CNT@CoS?
2. For battery-type electrodes, instead of capacitance, capacity should be calculated.
3. Fig. 9. (d): Cycling stability should be continued for 5000 cycles at least.
4. Fig. 9(b): why the CV test was continued up to -0.2 V on the negative side for the scan rates of 5-20 mV/s. All the CV curves should be tested in the same potential range. Also, for devices, the CV test should be starting from 0 V.
5. Fig. 8. Equivalent circuit should be provided for EIS fitting.
6. More explanation is required for the EIS spectra of the electrodes.
Round 2
Reviewer 2 Report
The authors have addressed all the comments and the revised manuscript could be accepted.
Reviewer 3 Report
The manuscript can now be accepted for publication.